# Minimal Clinically Important Difference of Tinnitus Outcome Measurement Instruments—A Scoping Review

**DOI:** 10.3390/jcm12227117

**Published:** 2023-11-15

**Authors:** Berthold Langguth, Dirk De Ridder

**Affiliations:** 1Department of Psychiatry and Psychotherapy, Bezirksklinikum, University of Regensburg, 93053 Regensburg, Germany; 2Section of Neurosurgery, Department of Surgical Sciences, Dunedin School of Medicine, University of Otago, Dunedin 9054, New Zealand; dirk.deridder@otago.ac.nz

**Keywords:** minimal clinically important difference, minimal detectable change, patient reported outcome, anchor-based method, distribution-based method, clinical trial

## Abstract

Objective: Tinnitus assessment and outcome measurement are complex, as tinnitus is a purely subjective phenomenon. Instruments used for the outcome measurement of tinnitus in the context of clinical trials include self-report questionnaires, visual analogue or numeric rating scales and psychoacoustic measurements of tinnitus loudness. For the evaluation of therapeutic interventions, it is critical to know which changes in outcome measurement instruments can be considered as clinically relevant. For this purpose, the concept of the minimal clinically important difference (MCID) has been introduced. Study design: Here we performed a literature research in PubMed in order to identify for which tinnitus outcome measurements MCID criteria have been estimated and which of these estimates fulfil the current methodological standards and can thus be considered as established. Results: For most, but not all tinnitus outcome instruments, MCID calculations have been performed. The MCIDs for the Tinnitus Handicap Inventory (THI), the Tinnitus Questionnaire (TQ), the Tinnitus Functional Index (TFI) and visual analogue scales (VAS) vary considerably across studies. Psychoacoustic assessments of tinnitus such as loudness matching have not shown sufficient reliability and validity for the use as an outcome measurement. Conclusion: Future research should aim at the confirmation of the available estimates in large samples involving various therapeutic interventions and under the consideration of time intervals and baseline values. As a rule of thumb, an improvement of about 15% can be considered clinically meaningful, analogous to what has been seen in other entirely subjective pathologies like chronic pain.

## 1. Introduction

Tinnitus has been defined as the conscious awareness of a tonal or composite noise for which there is no identifiable corresponding external acoustic source. Tinnitus becomes tinnitus disorder when tinnitus is associated with emotional distress, cognitive dysfunction, and/or autonomic arousal, in other words suffering, leading to behavioral changes and functional disability [1].

### 1.1. Tinnitus Measurement

The quantification of tinnitus and tinnitus-associated suffering is challenging as both the sound percept and the associated suffering are of a primarily subjective nature. In recent years, many efforts have been made to develop and validate instruments for the measurement of tinnitus [2]. In principle, these approaches can be differentiated in two groups: instruments that aim at characterizing the loudness of the tinnitus percept and those that aim to quantify tinnitus-related suffering, in keeping with the definition of tinnitus and tinnitus disorder. For the assessment of tinnitus suffering, several validated questionnaires are used. It has been recommended that the primary outcome for clinical treatment trials should be validated questionnaires [3]. More than 70 tinnitus questionnaires exist, many of which are non-validated [4]. Some widely used questionnaires differ in the selection of domains that might be negatively affected by tinnitus in everyday life (see Table 1). Some tinnitus questionnaires have no or few tinnitus-specific questions, such as the THI, TRQ and THQ, respectively (see Table 1). Some questionnaires focus predominantly on emotional or psychological aspects. For example, the TRQ has 77% of questions related to emotional and psychological aspects, followed by the THI (66%), TQ (47%), THQ (44%) and TFI (36%). The TFI and TQ are the most evenly distributed questionnaires, covering most domains evenly. The TQ is a very similar construct, but has slightly more questions related to emotional impact. The TRQ almost uniquely addresses the emotional response (77% of questions) and impact on lifestyle (19%), with some sleep-related questions (4%). The THI is similar to the TRQ but adds some information of auditory perception (4%) and general health (4%). Thus, depending on what aspect of tinnitus-associated suffering one wants to evaluate, a different questionnaire can be selected.

The quantification of suffering or burden can also be assessed by visual analogue scales (VAS), numeric rating scales (NRS) or verbal rating scales for annoyance and distress. When measured repetitively within a short time frame (3 days), the VAS annoyance and VAS distress values show very high agreement (93% and 96%, respectively [5]), and at longer time differences (14 days), the test–retest reliability of VAS annoyance is 0.79 [6]. The assessment of tinnitus complaints by a questionnaire can be complemented by the assessment of tinnitus loudness by either psychophysical measurements or by a numeric rating scale. NRS ‘subjective’ loudness measurements can detect changes much better than what may be considered more ‘objective’ psychoacoustic measures, and VAS loudness has a good test–retest reliability of 0.8 [6]. These recommendations are reflected using different tinnitus measurements in the literature, as evidenced by review [2].

### 1.2. Minimal Clinically Important Difference (MCID)

When assessment instruments are used for the investigation of therapeutic interventions, it is important to determine what the minimal change of the score of a given instrument is that reflects a clinically relevant or meaningful improvement [7]. This is not only important for assessing diagnostic and therapeutic management in routine clinical practice, but also for planning and evaluating the clinical trials of new pharmaceuticals and medical devices [8,9]. The clearance of therapeutic products by health authorities as well as labeling and promotional claims must be based on evidence for clinical efficacy. This in turn requires the determination of a minimal clinically meaningful, i.e., important, difference (MCID).

### 1.3. Patient Reported Outcomes (PROs)

Methodological requirements for PRO instruments include a clear conceptual framework, evidence supporting content validity, and acceptable psychometric qualities (e.g., reliability and validity [9]). The PRO measures must also be validated with respect to responsiveness or sensitivity to changes in clinical status to be most useful as effectiveness endpoints in clinical trials. Responsiveness is an aspect of construct validity and is determined by evaluating the relationship between changes in clinical and other endpoints and changes in the PRO scores over time, or based on the application of a treatment of known and demonstrated efficacy, in either observational studies or in clinical trials [9,10].

Demonstrating responsiveness is necessary, but additional information is needed to determine the MCID for a PRO measure. The MCID has been defined as the smallest change in a PRO measure that is perceived by patients as beneficial or that would result in a change in treatment [10,11]. There is no one unique method of calculating the MCID. There exist a number (at least 10 [12]) of methods for calculating MCID, and these can be broadly divided in anchor-based and distribution-based methods [13,14,15]. These different methods of calculating the MCID can result in important differences in what is to be considered clinically meaningful. For example, in pain, which like tinnitus is an entirely subjective experience and for which a NRS painfulness quantification and suffering questionnaire quantification exists, a comparison has been made by using different MCID computations [16]. For the NRS, on a scale from 0 to 10 for painfulness, the MCID differs between 0.9 and 2.7 depending on the method of calculating the MCID, which results in a dramatic difference in what is considered clinically relevant. To put it into perspective, a 0.9 NRS difference on a scale of 0 to 10 equals an improvement in painfulness of 9% of the total scale range. This is in sharp contrast to an improvement of 2.7 points, which translates to 27% improvement. An even more extreme example is the MCID for the pain catastrophizing scale, which has a range of 0 to 52. The most lenient MCID suggests that a change of 1.9 points (7% of the total scale range) is sufficient to have a clinically meaningful improvement, but the most stringent calculation requires a 13.6 point (26.2% of the total scale range) improvement [16].

### 1.4. Anchor-Based Approaches

Anchor-based approaches use an external marker of change, termed the anchor, to identify the occurrence of change in the target concept of interest; for example, whether a relevant change in pain, function, or quality of life occurred. These anchors become the critical part of the analysis.

An MCID value is influenced by the choice of the anchor (i.e., criterion measure), whose perspective is used on the anchor, and what the direction of change is in relation to the anchor. Typically, the MCID depends primarily on the patient’s perspective [17], i.e., whether the patient subjectively perceives a clinical improvement, and this is particularly relevant for determining the MCID in PROs that are the standard for outcome measurement in entirely subjective conditions such as pain or tinnitus. It has been proposed that MCID calculations should be based on PROs, such as the Clinical Global Impression of Improvement Scale (CGI-I) [18], as this indeed reflects the subjective improvement or worsening that the patient experiences resulting from a particular therapeutic intervention.

When using such an anchor-based approach, the individual rates the subjectively perceived change on the CGI-I between pre-treatment and post-treatment from the patient’s perspective. An a priori determination is made of what is to be considered an important change on this anchor, and after fielding it along with the questionnaires, the change score on the target questionnaire that relates to this threshold of important change on the anchor is calculated and reported as the MCID. Various thresholds and various statistical techniques can be applied for this calculation [19].

### 1.5. Distribution-Based Approaches

Distribution-based approaches use internal quantifications of statistical variability in the sample and magnitude of effect as a proxy to MCID quantification in the target measure of interest [11,20,21]. Typically, they include one standard error of measurement (Sem) [19], 0.5 standard deviation (SD) [22], or change equivalent to a 0.2 or 0.5 effect size. Other related statistical methods for estimating measurement errors (such as Guyatt’s responsiveness index, the reliable change index, standard response difference [SRD], Bland and Altman’s limits of agreement, and minimal/smallest detectable change [MDC/SDC]; [21,22,23,24]) are not recommended as distribution-based approaches to MCID approximation [19]. Moreover, distributional approaches alone are not encouraged as a sole method for MCID determination because of the lack of subjective valuation from the patient’s perspective of the importance of that change.

Frequently, the Bland–Altman plot is used to visually represent the measurement error of an instrument. The Bland–Altman plot is based around three variables: the mean systematic difference between test and retest scores and the upper and lower limits of agreement, which span 95% of observations, assuming a normal distribution [13,22]. These variables are integrated into a scatter plot where the difference between test and retest values is plotted on the Y-axis and the average of the test and retest values is plotted on the X-axis.

The range of ±2 SD around the mean value indicates the range where 95% of values can be found if the values are randomly distributed. In other words, if a value is outside of this range, it is significantly different from the mean value at a significance level of *p* = 0.05. Thus, this 95% confidence interval illustrates the amount of an individual change in the instrument’s score that has to be achieved to differentiate it from measurement error [20]. These metrics can be interpreted as the MDC or SDC of an instrument, but should not be used as a replacement for the MCID, as it reflects primarily an estimator of the statistical measurement error of an instrument [19], but does not verify concepts of subjective important or meaningful change [24,25,26].

### 1.6. State of the Art for Determining MCID

The recommended approach is to estimate the MCID based on several anchor-based methods, with relevant clinical or patient-based indicators, and to examine various distribution-based estimates (i.e., effect size, standardized response mean, standard error of measurement) as supportive information, and then to triangulate on a single value or small range of values for the MCID [18]. This procedure ensures that a change in scores over time is meaningful for the patient (anchor-based) and that the measurement error of the instrument is small enough (distribution, i.e., statistics-based) to detect this meaningful change. In addition, confidence in a specific MCID value evolves over time and is confirmed by additional research evidence, including clinical trial experience [14]. This approach is also reflected by the current guidance of the Food and Drug Administration (https://www.fda.gov/media/77832/download (accessed on 29 October 2023)):


*“The empiric evidence for any responder definition is derived using anchor-based methods. Anchor-based methods explore the associations between the targeted concept of the PRO instrument and the concept measured by the anchors.*



*The difference in the PRO score for persons who rate their condition the same and better or worse can be used to define responders to treatment.*



*Distribution-based methods for determining clinical significance of particular score changes should be considered as supportive and are not appropriate as the sole basis for determining a responder definition”.*


This is a very important statement from the FDA, as it puts quantification of a subjective state (tinnitus, pain, …) back into the clinical realm, and only uses statistics to make sure that what we use a measurement scale can be trusted from a statistical point of view.

The objective of this scoping review is to summarize the existing literature about MCIDs for the most commonly used tinnitus outcome measurements, in order to identify whether converging evidence exists for MCIDs of the various questionnaires or whether further research is needed to establish such evidence.

## 2. Methods

In this scoping review we followed the PRISMA guidelines to determine for which of the most frequently used tinnitus outcome measurements the minimal relevant change has been estimated. A systematic literature research was performed in PubMed Central (https://www.ncbi.nlm.nih.gov/pubmed) on 6 January 2023. We included all outcome measurements in our search, which were used in at least 15 studies according to a recent review [2] (tinnitus loudness match; numeric rating scales for tinnitus loudness and annoyance; visual analogue scales for tinnitus loudness and annoyance; Tinnitus Handicap Inventory (THI); Tinnitus Questionnaire (TQ); Tinnitus Functional Index (TFI)). For the literature search we used the search terms [*Name of the Outcome Measurement Instrument*] and ([MCID] or [MID] or [minimal clinically important difference] or [minimal important difference] or [minimal detectable change] or [MDC] or [smallest detectable change] or [SDC]). In addition, the original publications of the different instruments were screened for any information related to the minimal relevant change. In all identified publications, both authors critically reviewed which measures of MCID were determined, which method was used and on which data the estimations were based. The resulting information is displayed in Table 2.

## 3. Results

The systematic literature research identified 35 studies. After the removal of duplicates [10] and studies that were not relevant for our purposes [14], 11 studies remained for data extraction, which can be found in Table 2.

Estimations of the MDC or MCID have been performed for most, but not all, of the investigated instruments. Moreover, different approaches have been used. Some analyses primarily determined the MDC by using a distribution-based method that is derived from test–retest reliability measurements [37]. Others determined the MCID by combining anchor-based and distribution-based approaches. An overview of the MDC and MCID for the different instruments including the used methods is given in Table 2.

### 3.1. Minimal Detectable Change (MDC) for Tinnitus Outcome Measurement Instruments

The MDC has been determined and illustrated by Bland–Altman plots for the THI [27], the TFI [32,33] and for tinnitus loudness matching and tinnitus loudness rating [36]. For the THI, there are also anchor-based estimates for the MCID that reveal clearly different results (see Table 2). In contrast, for tinnitus loudness matching, only the MDC was determined, as the anchor-based approach revealed that the subjectively perceived improvement (CGI-I) did not correlate with the change in tinnitus loudness matching [36]. This finding indicates that the tinnitus loudness match is not suitable as an outcome measurement for tinnitus.

### 3.2. Minimal Clinically Important Difference (MCID) for Tinnitus Outcome Measurement Instruments

For the determination of the minimal clinically important difference (MCID) in PROs, a combination of anchor-based and distribution-based methods was recommended [11]. This recommended method has been used for the Tinnitus Handicap Inventory (THI) [28], the Tinnitus Questionnaire (TQ) [29,30], and the Tinnitus Functional index (TFI) [31,32,35], as well as for numeric scales of tinnitus loudness [6,36]. As the anchor, either the patient-rated version of the clinical global impression-improvement scale (CGI-I) or the global perception of change (GPC) scale was used. In both scales, the subjectively perceived improvement from pre-treatment to post-treatment is rated by the patient. It is important to note that the CGI-I consists of a 7-point scale (3 = ‘much improved’, 2 = ‘moderately improved’, 1 = ‘slightly improved’, 0 = ‘no change’, −1 = ‘slightly worse’, −2 = ‘moderately worse’ to −3 = ‘much worse’). In some of the studies, the anchor for the MCID was “slightly improved”; in others, the anchor was “moderately improved”, and in some analyses CGI response categories were collapsed from the 7-point scale to a 3-point scale of ‘improved’ (much to slightly improved), ‘no change’, and ‘worsened’ (slightly to much worse).

There is some variation in the use of distribution-based methods. Whereas most studies used the “half a standard deviation” method [22], some studies also used Intra-Class Correlations (ICC), limits of agreement (LoA) and SDC as supportive information to determine that the measurement error was small enough to observe meaningful changes in patients’ health status. For all questionnaires (THI, TQ, TFI), the anchor-based and distribution-based methods converged in the same range, enabling the estimation of MCID values. The MCID estimation by Zeman et al. [28] was the first MCID estimation of a tinnitus questionnaire, which followed the recommendation to combine anchor- and distribution-based methods. All further studies that determined MCID for other tinnitus outcome measurements followed this approach [6,28,29,30,31,32,35,36].

### 3.3. Tinnitus Handicap Inventory (THI)

The THI consists of 25 items with 3 answer options that can be scored with 0, 2 or 4 points, respectively, resulting in a total score of 100. When Newman et al. developed the THI [27], they assessed its psychometric properties in a sample of 29 patients, resulting in a minimal detectable change of 20 points. Zeman et al. analyzed pre- and post-treatment data from 210 patients, who underwent various therapeutic interventions. The interval between pre- and post-treatment assessments was 4 weeks for about half of the patients and 12 weeks for the other half. ‘Slight improvement’ in the CGI-I was used as the anchor, and the link between CGI-I and the change in the THI score (pre-treatment versus post-treatment) was investigated by the method of equipercentile linking and by calculating Cohen’s effect sizes. The results from the anchor-based approach were then complemented by a distribution-based method, specifically the half of a standard deviation [22]. The equipercentile-linking approach suggested an improvement in the THI by six points or more as the best separation between ‘no change’ and ‘slight improvement’ in the CGI-I.

Calculating the effect sizes for the two CGI-I groups of ‘no change’ and “’minimally better’ revealed a Cohen’s d = 0.26 (95% CI, 0.03–0.46; ‘no change’) and Cohen’s d = 0.74 (95% CI, 0.42–1.05; “minimally better”), respectively, with the confidence intervals overlapping only slightly. An effect size of 0.5 was determined as the appropriate cut-off point, as d = 0.5 was outside of the 95% CI of the ‘no change’ group. These results were then compared with the pooled standard deviation of all studies (SD_diff_ = 13.8). The effect size of 0.5 corresponded exactly with the half standard deviation [22] resulting in a cut-off value of 7 points.

### 3.4. Tinnitus Questionnaire (TQ)

The Tinnitus Questionnaire (TQ) is a validated and commonly used instrument for the assessment of tinnitus severity used in many clinical studies consisting of 52 questions with 3 answer options (0, 1 or 2 points). Factor analysis of the German translation of the TQ revealed that the dimensions of emotional and cognitive distress, intrusiveness, auditory perceptual difficulties, sleep disturbances, and somatic complaints can be differentiated. This validation also resulted in a different scoring system, which is now most widely used. In this scoring system, some items are not used at all and others load in two factors, resulting in a maximum score of 84 points.

Adamchic et al. pooled data from 757 patients with chronic tinnitus from the TRI database and a sound treatment study [29]. An anchor-based approach using the clinical global impression (CGI) scale and distributional approaches were used to estimate MCID. TQ change scores and CGI ratings correlated relatively well (r = 0.52, *p* < 0.05). Using this anchor-based method, a difference of −6.65 points on the TQ was associated with a minimally better score on the CGI, and a minimally worse CGI was linked to +2.72 points on the TQ [29]. According to a distribution approach, for the receiver operating characteristic (ROC) method, the MCID for improvement was −5 points and for deterioration +1 points [29], yielding comparable results in identifying MCIDs.

Hall et al. assessed pre- and post-intervention scores of 202 patients with tinnitus undergoing music therapy. Following recommended standards, multiple estimates were computed using anchor- and distribution-based statistical methods. The six estimates calculated for the TQ MCID ranged from 5 to 21 points in TQ change score from pre- to post- intervention. The authors advised an MCID of 12 points for the TQ [30]. The difference between a MCID of 5 as suggested by Adamchic et al. [29] and a MCID of 12 as suggested by Hall et al. [30] is considerable. Therefore, further research is required to reveal a reliable estimate for the TQ. Yet, as mentioned above, these differences are not unique to tinnitus, but also exist in the pain literature.

Recently, the MCID was estimated for the Mandarin version of the TQ (37 items with a total score of 74; not included in Table 2) [38]. The MCIDs of the Mandarin TQ were also established via anchor-based and distribution-based methods in 115 patients. The MCIDs for the change in total Mandarin TQ ranged from 6.29 to 19.00, with those for improvement from 1.09 to 22.75, and those for deterioration from 3.50 to 7.64. The authors selected an absolute value of 7.5 as the MCID for the Mandarin TQ score.

### 3.5. Tinnitus Functional Index (TFI)

The tinnitus functional index (TFI) has been developed for use in both intake assessment and for measuring treatment-related changes in tinnitus [31]. Thus, it is the only questionnaire that has been specifically developed for assessing responsiveness to therapeutic interventions. Moreover, it was emphasized that the questionnaire provides comprehensive coverage of all important domains of negative tinnitus impact, including emotional/psychological impact, lifestyle impact, sleep, auditory perception, tinnitus-specific impact and general health impact (see Table 1). The validation procedure, however, resulted in 25 items and eight subscales for the domains ‘cognitive’, ‘auditory’, ‘intrusive’, ‘sleep’, ‘relaxation’, ‘quality of life’, ‘emotional’ and ‘sense of control’, i.e., two more subscales than are generally applied in tinnitus questionnaires. The answer options for each item range from 0 to 10. For the total score, the sum of all answers is divided by the number of answered items and then multiplied by 10. This procedure results in a score between 0 and 100.

Meikle used both an anchor and distribution approach in 155 patients treated for tinnitus [31]. The mean TFI change scores for the five change groups of the global perception of change variable at 3 and 6 months were used as the anchors. At 3 months, the score reduction for the ‘much to moderately improved’ group compared with the reduction for the ‘unchanged’ group was about 14 points, slightly larger than one half of the standard deviation (SD) observed for the initial scores. Likewise, after 6 months the TFI reduction for the slightly improved group compared with that of the unchanged group was about 17 points, again somewhat larger than one half the initial SD. Acknowledging the preliminary nature of their results from an observational study, the authors suggested that a reduction in TFI scores of around 13 points should be meaningful to patients.

Two further studies investigated the SDC by using distribution-based methods with divergent results [32,33]. Fackrell’s results suggest that the measurement error amounts to 22.4 [32], whereas it amounts to only 4.8 in the study from Chandra et al. [33]. Two more recent studies used a combination of anchor- and distribution- based methods and revealed 8.8 [34] and 14 [35], respectively, as estimates of the MCID. While Skarzynski investigated a sample of patients with otosclerosis receiving stapes surgery, Fackrell pooled observational data from patients receiving a large variety of treatments. The relatively large difference in the MCID estimate is presumably due to the fact that Skarzynski used the criterion ‘minimally improved’ as the anchor [34], whereas Fackrell pooled the data from the groups ‘minimally improved’, ‘much improved’ and ‘very much improved’ [35]. Moreover, Fackrell et al. demonstrated that both baseline scores and the interval between measurements have an impact on the MCID estimate.

### 3.6. NRS/VAS

Adamchic et al. analyzed data from a single-blind, randomized, placebo-controlled study of acoustic coordinated reset neuromodulation in patients with chronic tinnitus to assess the reliability, validity, and minimally clinically identifiable difference (MCID) of the VAS loudness and the VAS annoyance [6]. For the MCID, he combined distribution- and anchor-based methods. The VAS loudness and VAS annoyance were completed at screening at baseline, and at five visits during the 16 weeks of the clinical study. VAS loudness and VAS annoyance correlated well with the tinnitus questionnaire at all clinical visits (max r = 0.67, *p* < 0.05). MCID estimates determined with different methods (CGI-I; ROC, effect size, SEM) clustered between 10 and 15 points [6].

Hall et al. also used distribution- and anchor-based methods when she analyzed data from 76 patients participating in a randomized, double-blind, placebo-controlled pharmacological phase IIa trial [36]. Tinnitus loudness was rated on a Likert scale (range 0–10) at screening at baseline and at study end after 28 days of treatment. The mean NRS score of the ‘slightly improved’ group was about 1 point lower than that of the ‘no change’ group, and the mean score of the ‘much to moderately improved’ group was about 2.4 points lower than that of the ‘no change’ group. However, as the limits of agreement (LoA) analysis revealed that a change score up to 2.61 may be attributed to the measurement error, the authors suggested an MCID of 3.5 points on the Likert Scale. Interestingly, this is the only paper in which results from anchor-based and distribution-based methods were added for estimating the MCID. Analogous to the pain dataset [16], a difference of 3.5 points estimates that a difference of 32% is required for the patient to experience a clinical difference, which is likely an overestimation, and can only be applied if one accepts that the NRS has a 2.61 point measurement error. The inter-rater reliability of NRS test–retest in tinnitus is strong and estimated at 0.8 [39], which means that 64–81% of the data are considered reliable [40].

### 3.7. Loudness Match

In the same paper, Hall et al. also analyzed the results from loudness matching. An MCID could not be determined as changes in the tinnitus loudness matching did not correlate with CGI ratings. This was also noted in an earlier study [41].

## 4. Discussion

Knowledge about the minimal score reduction in an outcome measure that is clinically relevant, the MCID, is of utmost importance for the planning of clinical trials of new pharmaceuticals and medical devices, for the interpretation of their results and for the clearance of therapeutic products by health authorities. In this scoping review, we could identify several approaches to determine the MCID for the most frequently used tinnitus outcome measures [2]. We are aware that we might not have found all studies as our review was restricted to studies published in PubMed. However, as most peer reviewed publications are listed in PubMed, we think that the risk of missing high-quality peer reviewed publications by focusing on PubMed only is acceptable. A main finding was the high variability of the results from available studies. Various reasons may account for this variability.

First, not all studies used the recommended methodology, which is a combination of anchor- and distribution-based methods. Some studies only determined the SDC, which mainly reflects the measurement error of an instrument. However, even two studies that determined the SDC with the same method in two different samples (e.g., [32,33]), showed a large variability (22 versus 4.8). Differences between the two samples and the exact procedure of data collection together with a relatively small sample size in one study (40 patients [33]) may account for this difference. Yet, this discrepancy is not unique to tinnitus, and has been identified in larger research fields such as pain [16].

Second, studies that used a combination of distribution- and anchor-based methods varied in the anchor they chose. While all studies chose the clinical global impression-improvement (CGI-I) as the anchor, they varied in the item they used. Some used minimally improved, some used moderately or much improved and others pooled all improved groups. It is obvious that the choice of the CGI-I item determines the outcome of the analysis. Given the definition of the MCID, ‘minimally improved’ seems the most appropriate anchor.

Third, there are important confounding factors such as the interval between measurements and the baseline score. Fackrell et al. [35] analyzed these aspects in detail and could demonstrate that the score change of the TFI that is related with improvement in the CGI-I depends on the interval between measurements and the baseline score. Longer intervals and higher baseline scores are related to the need for higher score reductions to be perceived as improvements.

Fourth, the composition of the samples under study might play a role. What is considered as a clinically relevant change might differ between treatment-naïve and treatment-resistant patients, but also across cultures.

Fifth, presumably the type of intervention might also play a role. As shown in Table 1, the various questionnaires cover the different aspects of the tinnitus burden differently. Specific interventions might have different impacts on the various aspects of tinnitus. For example, CBT primarily focuses on reducing the cognitive and emotional burden, whereas hearing aids may improve communication and also reduce the perceived intensity of tinnitus. As the MCID should be universally valid independently of the specific treatment, it should be validated by data from various therapeutic interventions.

Further variability is added by the method of triangulating the results from distribution-based and anchor-based analyses. In most papers, the MCID is determined within the range where the values of both distribution and anchor-based methods converged. In one paper it was suggested that the measurement error should be added to the anchor-based value, resulting in a clearly larger MCID [36]. These different approaches might depend on whether the MCID is used in the evaluation of improvement for an individual patient or in the evaluation of a larger sample in a clinical trial. In the first case, the addition of the measurement error to the mean change score corresponding to minimal improvement in the CGI-I seems appropriate; in the second case it does not, since measurement errors should average out in a sufficiently large sample.

A further criterion for the relevance of a specific MCID is the acceptance of the value by the research community [4]. Currently, the THI and the TFI are the most frequently used questionnaires and the MCIDs for the THI ((−7; [28]) and the TFI (−13; [31]) can be considered as accepted by the research community. However, studies which use both scales for outcome measurement reveal a substantially different number of responders. This can be illustrated by the results of a recently published study, in which response rates of therapeutic interventions were determined based on the THI (MCID: −7), the TQ (MCID: −12) and the TFI (MCID: −13) [42]: the response rate varied between 19.1 (TFI), 19.7 (TQ) and 38.2 (THI). Given these huge differences, it would be highly desirable to reevaluate the MCID in further studies [1] by using identical methodology [2] in large samples, undergoing various interventions, and considering [3] intervals between assessment and [4] baseline values. The current lack of these data also represents the most important limitation of this review.

### A Rule of Thumb

As a rule of thumb, improvement of 15% or more of the total scale range seems to be clinically meaningful, both for visual/numeric rating scales and questionnaires. This value has recently been proposed as a general orientation for the MCID for patient reported outcomes (https://www.iqwig.de/presse/pressemitteilungen/pressemitteilungen-detailseite_27520.html (accessed on 29 October 2023)). Indeed, the obtained MCID, irrespective of the scale or questionnaire used, ranges by around 15% of the total score (see Table 2). This is similar to what is observed in MCID for other entirely subjective symptoms with similar clinical, pathophysiological and treatment approaches, such as pain (18% [7]) [43,44].

## 5. Conclusions

For most, but not all tinnitus outcome instruments, MCID calculations have been performed. The MCIDs for the Tinnitus Handicap Inventory (THI; 7points), Tinnitus Questionnaire (TQ, 12 points), Tinnitus Functional Index (TFI, 13 points) and visual analogue scales (VAS, 1.5 points) are currently used by the research community, but still must be considered as preliminary. As a rule of thumb, an improvement of 15% or more of the total scale range seems to be clinically meaningful, both for visual/numeric rating scales and questionnaires. Further research should aim at the confirmation of the available estimates in larger samples involving various therapeutic interventions using a more consistent methodology and considering baseline values and intervals between measurements.

## Figures and Tables

**Table 1 jcm-12-07117-t001:** Percentage of questions related to different impacts tinnitus has on different aspects of life.

	TFI	TQ	THQ	THI	TRQ
Emotion/psychology	36%	47%	44%	68%	77%
Lifestyle impact	12%	12%	22%	20%	19%
Sleep	12%	10%	4%	4%	4%
Auditory Perception	12%	8%	19%	4%	0%
Tinnitus specific	16%	10%	4%	0%	0%
Health	12%	13%	7%	4%	0%

TFI: Tinnitus functional index; TQ: Tinnitus Questionnaire; THQ: Tinnitus Handicap Questionnaire; THI: Tinnitus Handicap Inventory; TRQ: Tinnitus Reaction Questionnaire.

**Table 2 jcm-12-07117-t002:** Published reports about estimations of the minimal detectable change (MDC) and minimal clinically important difference (MCID) of the most frequently used tinnitus outcome measurements.

Outcome Measurement	Study	Sample Size	Sites	Intervention	Change Measurement	Method	Result (% of the Total Range)
Tinnitus Handicap Inventory	Newman et al., 1998 [27]	29	2	no	MDC	Distribution-based method (LoA, 95% CI)	20 (20%)
	Zeman et al., 2011 [28]	210	6	CBT, pharmacologic treatment, brain stimulation	MCID	Combination of anchor-based method (CGI-I, minimally improved) and distribution-based method (effect size, 0.5 SD)	7 (7%)
Tinnitus Questionnaire	Adamchic et al., 2012 [29]	757	7	Acoustic coordinated reset neuromodulation, brain stimulation, CBT and pharmacologic treatment		Combination of anchor-based method (CGI-I; minimally improved) and distribution-based method (0.5 SD, effect size, SEM)	5 (6%)
	Hall et al., 2018 [30]	202	1	Music therapy		Combination of anchor- based method (CGI-I) and distribution-based method (ICC, LoA, SDC)	12 (14%)
Tinnitus Functional Index	Meikle et al., 2012 [31]	155	3	Not standardized	MCID	Combination of anchor- based method (CGI-I; much or moderately improved) and distribution- based method (effect size, 0.5 SD)	13 (13%)
	Fackrell et al., 2016 [32]	294	2	No intervention	SDC	Distribution-based method (LoA, 95% CI, SEM)	22,4 (22%)
	Chandra et al., 2018 [33]	40	1	No intervention	SDC	Distribution-based method (SEM)	4.8 (5%)
	Skarzynski et al., 2018 [34]	95	1	Stapedotomy	MCID	Anchor-based method (CGI-I; minimally improved) (ROC)	8.8 (9%)
	Fackrell et al., 2022 [35]	255	12	Hearing aids, sound generators tinnitus maskers, sleep medications, relaxation training, yoga, mindfulness, and hypnosis	MCID	Combination of anchor-based method (CGI-I; improved) and distribution-based method (SEM, effect size, 0.5 SD)	14 (14%)
Visual analogue scales, Numeric Rating Scales	Adamchic et al., 2012 [6]	63	2	Acoustic coordinated reset neuromodulation,	MCID	Combination of anchor-based method (CGI-I; somewhat better) and distribution-based method (ROC, effect size, SEM)	15/100 (15%)
	Hall et al., 2017 [36]	76	15	Pharmacological treatment	MCID	Combination of anchor-based method (CGI-I; much improved) and distribution-based method (ICC, LoA, SDC)	3.5/10 (35%)
Tinnitus Loudness Matching	Hall et al., 2017 [36]	76	15		SDC *	Distribution-based method (LoA, 95% CI, SEM)	20.2 dB HL; 18.1 dB SL

Abbreviations: CGI-I: clinical global impression-improvement; SD: standard deviation; SEM: standard error of the mean; ICC: intra-class correlation; LoA: limits of agreement; SDC: smallest detectable change; ROC: receiver operating characteristic; CBT: cognitive behavioral therapy. * an anchor-based method (CGI-I) could not be applied as changes in loudness matching did not correlate with CGI-I.

## Data Availability

Data are contained within the article.

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
