# Peer review of "Minimal Clinically Important Difference of Tinnitus Outcome Measurement Instruments—A Scoping Review"

_jcm, 2023, doi:10.3390/jcm12227117_

Round 1

Reviewer 1 Report

Comments and Suggestions for Authors

Dear author(s),

I have reviewed the manuscript. My response is given in a point-by-point manner below.

Sincerely,

Minimal Clinically Important Difference (MCID) of Tinnitus Outcome Measurement Instruments – a scoping review

Title:

·         It’s good.

Abstract:

·         In the Study design section, it is better to add other research and database websites such as Google Scholar. 

·         In Result section, I recommended you to apply “Psychoacoustic assessments of tinnitus such as loudness matching” instead of “Psychoacoustic assessments of tinnitus loudness”.

·         In keywords, I think “approval” and “FDA” may be better to remove.

Main Text:

1. Introduction

·         Base on my opinion this section is comprehensive and consistent.

2. Methods

·         I recommend “adding other research databases such as Google Scholar” to the text. 

3. Results

3.1. Minimal Detectable Change (MDC) for tinnitus outcome measurement instruments

3.2. Minimal Clinically Important Difference (MCID) for tinnitus outcome measurement instruments

3.3. Tinnitus Handicap Inventory (THI)

3.4. Tinnitus Questionnaire (TQ)

3.5. Tinnitus Functional index (TFI)

3.6. NRS /VAS

3.7. Loudness match

·         I think “Result section” is comprehensive and can be effective in attracting the audience.

4. Discussion

·         Limiting the search to Pubmed can be one of the main limitations of this study. State your solution to solve this problem.

·         “Hearing aid is one of the subset of sound therapy techniques, which can improve the communication and also reduce the patient's perception of tinnitus”. I think apply this may be better than “hearing aids may rather improve communication”.

5. Conclusions

·         This section is comprehensive.

Author Response

We want to thank the reviewer for the constructive suggestions, which we answer below point by point

Minimal Clinically Important Difference (MCID) of Tinnitus Outcome Measurement Instruments – a scoping review

Title:

  • It’s good.

Abstract:

  • In the Study design section, it is better to add other research and database websites such as Google Scholar.

 Answer: We understand the concern of the reviewer. However, as most peer reviewed publications are listed in pub-med, we think that the risk of missing  high quality peer reviewed publications by focusing on pubmed only is acceptable

  • In Result section, I recommended you to apply “Psychoacoustic assessments of tinnitus such as loudness matching” instead of “Psychoacoustic assessments of tinnitus loudness”.

 Answer: done

  • In keywords, I think “approval” and “FDA” may be better to remove.

Answer: done 

Main Text:

  1. Introduction

  • Base on my opinion this section is comprehensive and consistent.

  1. Methods

  • I recommend “adding other research databases such as Google Scholar” to the text.

Answer: We understand the concern of the reviewer. However, as most peer reviewed publications are listed in pub-med, we think that the risk of missing  high quality peer reviewed publications by focusing on pubmed only is acceptable

  1. Results

3.1. Minimal Detectable Change (MDC) for tinnitus outcome measurement instruments

3.2. Minimal Clinically Important Difference (MCID) for tinnitus outcome measurement instruments

3.3. Tinnitus Handicap Inventory (THI)

3.4. Tinnitus Questionnaire (TQ)

3.5. Tinnitus Functional index (TFI)

3.6. NRS /VAS

3.7. Loudness match

  • I think “Result section” is comprehensive and can be effective in attracting the audience.

  1. Discussion

  • Limiting the search to Pubmed can be one of the main limitations of this study. State your solution to solve this problem.

Answer: we added the following explanation: “However, as most peer reviewed publications are listed in pub-med, we think that the risk of missing  high quality peer reviewed publications by focusing on pubmed only is acceptable”

  • “Hearing aid is one of the subset of sound therapy techniques, which can improve the communication and also reduce the patient's perception of tinnitus”. I think apply this may be better than “hearing aids may rather improve communication”.

  Answer: done

  1. Conclusions

  • This section is comprehensive.

Reviewer 2 Report

Comments and Suggestions for Authors

Thank you for the nice scoping review. Indeed as you have pointed out, different therapeutic interventions and interval of assessment make it heterogenous and hence difficult for establishing MCID agreement. 

Just some minor edits

Line 113 and 115

I believe you meant 1.9 points and 3.7% in line 113 and 26.2% in line 115

under Results, since you mentioned using PRISMA, please provide a PRISMA flow diagram to make it easier for us to see the methodology of scoping review according to PRISMA guidelines.

Author Response

We want to thank the reviewer for the constructive suggestions, which we answer below point by point

Thank you for the nice scoping review. Indeed as you have pointed out, different therapeutic interventions and interval of assessment make it heterogenous and hence difficult for establishing MCID agreement.

 Answer: Thank you

Just some minor edits

Line 113 and 115

I believe you meant 1.9 points and 3.7% in line 113 and 26.2% in line 115

Answer: done

under Results, since you mentioned using PRISMA, please provide a PRISMA flow diagram to make it easier for us to see the methodology of scoping review according to PRISMA guidelines.

Answer: we added a PRISMA Checklist in supplementary materials

Reviewer 3 Report

Comments and Suggestions for Authors

The topic is very interesting. Outcome measurement is very important part of assessment of effectiveness of a treatment or intervention, especially in defining of a such subjective feeling as tinnitus.  Several tinnitus specific instruments are used in clinical practice while the psychometric properties differs significantly. The issue of minimal clinically important difference needs to be systemized and well defined. The presented scopus review is well organized and comprehensive. The literature search has been done correctely and the sources are investigated thoroughly.

Just only comment on the manuscript is about the use of abbreviations. It is sometimes not consistent, please check them carefully. For example,

Lines 66, 71 – NRS – numeric rating scale?

Lines 65, 67 – VAS – visual analog scale (here the abbreviation is more close to the full form and more recognizable then first one)

Lines 146, 208-209, 257, 356 - MDC, SDC – first appearence of full form and abbreviation, then repeat use of full form and abbreviation

Lines 76, 84, 97, 403, 407 - MCID – full form is used several times

Line 371 – coordinated reset (CR) - the abbreviation is not used further

It is better to use full forms only in headings

Author Response

The topic is very interesting. Outcome measurement is very important part of assessment of effectiveness of a treatment or intervention, especially in defining of a such subjective feeling as tinnitus.  Several tinnitus specific instruments are used in clinical practice while the psychometric properties differs significantly. The issue of minimal clinically important difference needs to be systemized and well defined. The presented scopus review is well organized and comprehensive. The literature search has been done correctely and the sources are investigated thoroughly.

Just only comment on the manuscript is about the use of abbreviations. It is sometimes not consistent, please check them carefully. For example,

Lines 66, 71 – NRS – numeric rating scale?

Lines 65, 67 – VAS – visual analog scale (here the abbreviation is more close to the full form and more recognizable then first one)

Lines 146, 208-209, 257, 356 - MDC, SDC – first appearence of full form and abbreviation, then repeat use of full form and abbreviation

Lines 76, 84, 97, 403, 407 - MCID – full form is used several times

Line 371 – coordinated reset (CR) - the abbreviation is not used further

It is better to use full forms only in headings

Answer: thank you for this comment; we critically revised the manuscript with respect to a more consistent use of abbreviations; we only opted for keeping the abbreviations also in the headings